# Association between Fitness Level and Physical Match Demands of Professional Female Football Referees

**DOI:** 10.3390/ijerph191710720

**Published:** 2022-08-28

**Authors:** María Luisa Martín Sánchez, José M. Oliva-Lozano, Jorge García-Unanue, Peter Krustrup, Jose Luis Felipe, Víctor Moreno-Pérez, Leonor Gallardo, Javier Sánchez-Sánchez

**Affiliations:** 1Faculty of Sport Sciences, Universidad Europea de Madrid, Calle Tajo, s/n, Villaviciosa de Odón, 28670 Madrid, Spain; 2Health Research Centre, University of Almería, 04120 Almería, Spain; 3IGOID Research Group, University of Castilla-La Mancha, 45004 Toledo, Spain; 4Faculty of Health Science, University of Southern Denmark, 5230 Odense, Denmark; 5Center for Translational Research in Physiotherapy, Department of Pathology and Surgery, Miguel Hernandez University of Elche, 03202 San Joan, Spain

**Keywords:** load monitoring, officials, tracking systems, soccer, performance

## Abstract

The aim of this study was to examine the physical demands for elite female referees during competitive matches and to evaluate the relationship between match performance and fitness levels. Seventeen female elite field referees were fitness-tested (29.0 ± 5.2 (SD) years, 163.8 ± 6.7 cm, 54.0 ± 5.1 kg, Yo-Yo Intermittent Recovery Test Level 1 (YYIR1) performance 1610 ± 319 m) and were analysed during a total of 187 football matches. Global positioning systems (GPS) were used in this research. The physical demands were significantly lower (*p* < 0.05) in the second half of matches compared to the first half. Regarding the acceleration-related variables, the female referees revealed a higher number of high-intensity actions in the first half compared to the second half. However, higher low-intensity demands were observed for the females referees in the second half compared to the first half in the total number of accelerations (ACC) (~70.48 n; ES = 0.61; *p* < 0.05) and decelerations (DEC) (~71.11 n; ES = 0.62; *p* < 0.05); total number of ACC in zone 1 (~85.27 n; ES = 0.70; *p* < 0.05) and DEC in zone 1 (Z1) (~83.98 n; ES = 0.71; *p* < 0.05); and distance covered accelerating and decelerating in Z1. The distance covered in YYIR1 and the performance in the repeated sprint ability test correlated with the physical demands during matches for female football referees (*p* < 0.05). In conclusion, this study described the physical performance profile of female football referees and differences between the first and second halves of matches were observed. The results of this study revealed positive correlations between intermittent exercise fitness levels, the repeated sprint ability performance and match performance in female football referees, and thus this information can be implemented in their training plan designs.

## 1. Introduction

In order to obtain a competitive advantage, clubs and organizations such as Fédération Internationale de Football Association (FIFA) [1] are investing time and financial resources in technologies that can quantify the characteristics of training and match demands in a valid and reliable manner [2]. Specifically, several previous studies have analysed the suitability and reliability of global positioning systems (GPS) with football referees [3,4,5].

Currently, most football science studies are based on the physical performance profiles of athletes while fewer studies have focused on referees [6,7]. In elite football, a match cannot take place without the participation of the referees, but their profiles have not been widely investigated [8]. The female participation in sports has increased; however, there are not studies about the relationship between the fitness level and the physical match demands in female football referees [9]. The physical demands of football referees are partly associated with the performance of the players, so their physical fitness, high-intensity actions and fatigue must be examined [10,11].

Elite female football players cover an average of between 9 and 10 km during matches, with approximately ~15% of this distance covered at high running speeds [11]. In this regard, during an elite football match, a football referee can cover a distance of ~11.5 km (range 9–14 km) [12,13,14], and more specifically, female football referees can cover a distance of ~10 km [9]. Previous studies found that female field referees had lower levels in performance compared to male referees [15].

Currently, the total of football matches in the same week has incremented exponentially, so players and referees experience congested calendars with lower resting time and, therefore, higher risk of injury [16]. From a practical perspective, quantifying the load experienced by female football referees is necessary for gaining a better understanding of match demands and their performance during the season [17]. The referee’s performance can be evaluated through the different battery of fitness tests to assess their football-related fitness levels [18]. Previous studies have examined the correlation between certain match performance variables and results of fitness tests in male international football referees [19], and the results revealed that a high sprint and cardiovascular fitness level could be relevant to the performance of male referees during matches. However, to the best of our knowledge, no studies have investigated the status of these fitness tests in elite female football referees.

Therefore, the main aim of this study was to analyse the association between the physical performance in a match and the fitness level of elite female football referees in order to specifically guide female football referees’ training.

## 2. Materials and Methods

### 2.1. Experimental Approach

This study followed a prospective observational design. A total of 187 football matches were analysed using wearable tracking systems, which recorded the physical match demands of female field referees.

### 2.2. Subjects

Seventeen female (age: 29.0 ± 5.2 years, height: 163.8 ± 6.7 cm and weight; 54.0 ± 5.1 kg) elite football referees participated in the study. Four referees were excluded due to problems with their performance tracking systems. All referees had more than 10 years of experience refereeing and a minimum of 4 years in the top Spanish football league. Female referees trained an average of four sessions per week (i.e., strength, endurance, and speed). They usually refereed one match during the week. As an inclusion criterion, referees had to be free of injuries (i.e., any physical complaint sustained by a referee resulting from a match or training that led to an absence of the next training session or match) [20]. Each referee was informed about the study, and they provided informed consent to participate. The project was approved by the Bioethics Committee for Clinical Research of Virgen de la Salud Hospital in Toledo (Ref.: 629; 17 February 2021). All subjects belonged to the Technical Committee of Referees (TCR). Real Federación Española de Fútbol and TCR authorized this investigation.

### 2.3. Procedure

One fitness test for the female football referees was the repeated sprint ability test (RSA) and consisted of six 40 m sprints with one minute of recovery [9]. This sprint of 40 m was registered with a system of four pairs of photocells (Microgate, Bolzano, Italy) placed at 0, 5, 25, and 30 m, which collected time with a sensibility of 0.001 s [21], and the time measurements were in four distance-intervals (0–5; 5–25; 25–30; 0–30 m) [22]. This test is related to the ability to perform repeated high-intensity actions and to evaluate physiological parameters such as maximal oxygen uptake, among others [23]. Total time (RSA_TT_), the best time (RSA_BEST_), the percentage of best time (% RSA_BEST_), and the difference between the best and worst sprint during the RSA test (% RSA_DIFF_) were calculated [21]. In addition, the female football referees performed a YYIR1 to determine their total distance achieved in the test [24]. The female referees had to run 2 × 20 m out and back with a progressive speed set by the rhythm of the audios and a 10 s active break consisting of 2 × 5 m jogging. Once the referee did not reach the line in time, the distance covered was recorded and the result [1] was used to determine their aerobic capacity.

Data were collected from 187 matches during the 2020–2021 season. Matches took place in different football stadiums with similar dimensions according to the FIFA requirements (110 × 70 m). Referees’ physical match demands were monitored using WIMU PRO^TM^ (RealTrack System SL, Almería, Spain). Each device collected data at 10 Hz and had its own internal microprocessor with a high-speed USB interface to record, store and upload data with an authorized computer protocol [25]. All devices were activated 30 min before data collection to allow for the acquisition of satellite signals and synchronization of the GPS clock with the satellite’s atomic clock [26]. All referees completed a standard warm-up before the match for 15 min. However, these data were excluded. Only data collected during the first and second half were considered for the analysis. 

The physical demands were represented as external load variables, which were downloaded from the intervals pro report on SPro (Realtrack Systems SL, Almería, Spain). Specifically, the following variables were included: total distance (m), explosive distance (total distance covered with accelerations (ACC) above 1.12 m/s^2^; m), high-intensity breaking distance (HIBD: distance decelerating > 2 m/s^2^), total number of sprints (n), sprint distance (m), high-speed running distance (HSRD in m > 21 km/h), high-speed running actions (HSRA, n) [27], maximal speed (km/h), distance covered in different speed zones (Z1: 0–6 km/h; Z2: 6–12 km/h; Z3: 12–18 km/h; Z4: 18–21 km/h; Z5: 21–24 km/h; and Z6: >24 km/h), total number of ACC (*n*), total number of decelerations (DEC) (*n*), maximal acceleration (ACC_MAX_: m/s^2^), maximal deceleration (DEC_MAX_: m/s^2^), acceleration/deceleration ratio (Acc/Dec), mean acceleration (ACC_MEAN_: m/s^2^), mean deceleration (DEC_MEAN_: m/s^2^), total number of accelerations and distance covered accelerating by zones (Z1: 0–1 m/s^2^; Z2: 1–2 m/s^2^; Z3: 2–3 m/s^2^; Z4: >3 m/s^2^), and total number of decelerations and distance covered decelerating by zones (Z1: 0–1 m/s^2^; Z2: 1–2 m/s^2^; Z3: 2–3 m/s^2^; Z4: >3 m/s^2^) [28]. 

### 2.4. Statistical Analysis

The descriptive statistics were presented as mean ± standard deviations. Firstly, a Kolmogorov–Smirnov test was used to test the normality of the data (*p* > 0.05). In order to explore the differences in physical match demands between the first and second halves, a paired samples *t*-Test was performed. A repeated measures ANOVA for six different times was also performed to analyse the difference between the time obtained in the second to sixth sprint of RSA and the first one. Effect size (ES) was also calculated by Cohen’s d and defined as follows: trivial (ES < 0.19), small (ES = 0.20–0.49), medium (ES = 0.50–0.79), and large (ES > 0.8) [29]. Finally, a bivariate Pearson correlation was used to analyse the relationship between the physical demands and results obtained in the performance tests. The level of significance was set at *p* < 0.05 and all analyses were performed using the SPSS package (v24, SPSS Inc., Chicago, IL, USA).

## 3. Results

Figure 1 shows the time performance of the RSA test values. Statistical analysis reported a RSA_TT_ (35.15 ± 1.36 s) and a RSA_BEST_ (5.78 ± 0.24 s). RSA fatigue analysis revealed a % RSA_BEST_ (1.39 ± 0.72 s) and % RSA_DIFF_ (2.97 ± 1.59 s). Fatigue analysis during the RSA test showed reduced performance in sprint six compared to sprint one (+0.06 s; CI 95%: 0.01 to 0.11 s; ES: 0.26). Finally, the female football referees covered an average of 1610 ± 319 m in the Yo-Yo Intermittent Recovery level 1 test.

The analysis of physical demands in the first and second half for female field referees are outlined in Table 1 and Figure 2 for distance- and speed-related variables. The physical demands were significantly lower in the second half compared to first half in the total distance covered (~69.48 m; ES = 0.17; *p* < 0.05), explosive distance (~43.80 m; ES = 0.36; *p* < 0.05), maximal speed (~0.42 km/h; ES = 0.20; *p* < 0.05), distance covered in Z2 (~80.20 m; ES = 0.30; *p* < 0.05), distance covered in Z3 (~57.23 m; ES = 0.21; *p* < 0.05), and distance covered in Z6 (~1.02 m; ES = 0.04; *p* < 0.05). In addition, the female referees showed higher results in the second half than the first half in HIBD (~22.96 m; ES = 0.72; *p* < 0.05) and distance covered in Z1 (~78.94 m; ES = 0.27; *p* < 0.05). 

The physical demands related to the acceleration and deceleration variables are shown in Table 2. The results showed greater physical demands in the first half compared to the second half in ACC_MAX_ (~0.17 m/s^2^; ES = 0.23; *p* < 0.05), DEC_MAX_ (~0.25 m/s^2^; ES = 0.33; *p* < 0.05), ACC/DEC ratio (~1.22 m/s^2^; ES = 0.19; *p* < 0.05), total of ACC in Z2 (~7.89 n; ES = 0.20; *p* < 0.05), total of ACC in Z3 (5.37 n; ES = 0.30; *p* < 0.05), total of ACC in Z4 (~1.54 n; ES = 0.21; *p* < 0.05), distance covered accelerating in Z2 (~2.38 m; ES = 0.02; *p* < 0.05), distance covered accelerating in Z3 (~27.59 m; ES = 0.37; *p* < 0.05), distance covered accelerating in Z4 (~15.21 m; ES = 0.34; *p* < 0.05), total of DEC in Z3 (~6.61 n; ES = 0.39; *p* < 0.05), total of DEC in Z4 (~2.76 n; ES = 0.33; *p* < 0.05), distance covered decelerating in Z3 (~34.33 m; ES = 0.26; *p* < 0.05), and distance covered decelerating in Z4 (17.90 n; ES = 0.26; *p* < 0.05). 

However, greater physical demands were observed for the females referees in the second half compared to the first half in the total of ACC (~70.48 n; ES = 0.61; *p* < 0.05), the total of ACC in Z1 (~85.27 n; ES = 0.70; *p* < 0.05), distance covered accelerating in Z1 (~31.97 m; ES = 0.21; *p* < 0.05), total of DEC (~71.11 n; ES = 0.62; *p* < 0.05), total of DEC in Z1 (~83.98 n; ES = 0.71; *p* < 0.05) and distance covered decelerating in Z1 (~39.27 m; ES = 0.24; *p* < 0.05).

Pearson’s correlation (Table 3) showed a significant correlation between sprint distance and result in RSA_TT_ (r = −0.66; *p* < 0.01) and RSA_BEST_ (r = −0.67 m; *p* < 0.01). RSA_TT_ (r = −0.50 m; *p* < 0.05) and RSA_BEST_ (r= −0.49; *p* < 0.05) had a significant correlation with values of match variables in HSRD. In addition, the female referees showed a significant correlation between RSA_TT_ (r = −0.63; *p* < 0.01), RSA_BEST_ (r = −0.63; *p* < 0.01), and maximal speed in the matches.

The results related to the acceleration and deceleration match variables showed a significant correlation between ACC in matches and RSA_TT_ (r = −0.55; *p* < 0.05), RSA_BEST_ (r = −0.51; *p* < 0.05); and the number of decelerations in the match and RSA_TT_ (r = −0.53 n; *p* < 0.05), RSA_BEST_ (r = −0.49; *p* < 0.05). 

The main significant correlations between match variables and the Yo-Yo Intermittent Recovery Test were shown in explosive distance (r = 0.47; *p* < 0.05), HIBD (r = 0.51; *p* < 0.05), ACC Z4 (r = 0.53 n; *p* < 0.05), DEC Z4 (r = 0.46; *p* < 0.05), and ACC Z4 (r = 0.54; *p* < 0.05).

There was no significant correlation between fatigue in the RSA test (% RSA_BEST_ and % RSA_DIFF_) and match variables (*p* > 0.05).

## 4. Discussion

The purpose of this study was to analyse the association between the physical performance in a match and the fitness level of elite female football referees. The main finding was that female Spanish referees experienced higher results in most of the analysed high-intensity variables in the first half of matches compared to the second half. To the best of the authors’ knowledge, this is the first study analysing the physical demands in female field referees from elite football leagues. On the other hand, the results of the fitness test showed a strong relationship with the physical demands in the match, therefore, it is very important to evaluate the fitness level of female football referees with the physical test in order to know their ability to perform in a match [26,30]. Previous analyses have also reported a relationship between the results in the RSA test and YYIR1 test and the match-related physical fitness of football referees [31,32,33]. These studies showed that male football referees that have higher cardiovascular capacity could cover higher distances at high speeds and intensity during the match [19]. The correlation analysis showed how a better performance in the RSA test was associated with higher sprint distances and peak match speeds like in previous studies [14,34]. On the other hand, a better performance in the YYR1 test was associated with more high-intensity ACC and DEC (number and distance) and higher explosive match distances achieved, similar to the associations found in male football referees [14,19]. However, there are no previous studies showing this correlation in female football referees. 

Our results in total distance (first half ~5007 m; second half ~4938 m) are lower in comparison with previous studies with male football referees from Brazilian first division (first half ~5219 m; second half ~5230 m) [13] or England first division (first half ~5832 m; second half ~5790 m) [35]. However, in total distance, these results are in line with other research about international female players’ performance, such as wide defender (~9892 m) [36].

The female field referees showed lower results in the second half in total distance, explosive distance, maximal speed and high-intensity speed zones. These results indicate that the referees’ performance is reduced in the second half. The lower performance evidenced in the second half of the football match could be related to fatigue [16] or contextual variables of the competition [37]. However, previous research has suggested that this may be because of game interruptions and effective playing time [38,39]. If we consider the different speed zones, other studies have not shown differences between the first and second half [40], but these results must be interpreted with caution because these authors used different speed zones. On the other hand, the results evidenced that there were higher results in the first half in some ACC and DEC zones, both total and peak, especially in the high-intensity actions. In terms of low-intensity activities, the female referees showed lower results in the first half in certain acceleration and deceleration zones, and in the acceleration/deceleration ratio for both total and peak. There are no studies that have evaluated the ACC and DEC profile in female football referees in this regard, but similar results can be observed in high-intensity actions between the female football referees of this research and national (non-professional) male referees [2,41] where these kinds of activities decrease in the second half.

This study reported the ACC and DEC profile of elite female football referees. These actions are very physically demanding [28] and can lead to increased fatigue, perceived exertion and risk of injury [37,42]. In this study, the ACC and DEC (total numbers or peak values) were analysed according to their intensity, as has been conducted in previous studies [19]. In addition, if we consider the results in these variables, these female referees had lower values than male football referees in Spanish First and Second Division [18]. In addition, this study is in line with other studies which found that field football referees had similar performance to football players [23,43]. Therefore, the results in HIBD and the low-intensity speed zone are consistent with other research. The differences between referees and players may be due to the fact that teams have the possibility to make substitutions to maintain a good performance during the match, while the referees cannot be replaced (except for injury), so these substitutions, fatigue and effective playing time can explain this lower female football referee’s performance [44,45].

The main limitation was the interruption of the season because of COVID-19 since matches had to be suspended. This situation caused a congested calendar with more than one match per week. Furthermore, no internal load variables were included in the study, and these variables may be of interest for coaches since they may explain how female referees respond psycho-physiologically to the load. In addition, future research should consider the analysis of both match and training demands. Moreover, from a practical standpoint, the analysis of the physical demands of female referees officiating male matches is necessary to have a better understanding of match demands.

## 5. Conclusions

In conclusion, differences between the first and second halves of matches were observed in physical demands required by female field referees. The results showed positive correlations between the fitness levels and the physical demands during match play competition in the female football referees. The fast evolution and professionalization of female football and the use of technology to control match loads can help coaches design training strategies to decrease the risk of injury and optimize performance in female football referees. 

### Practical Application

Most of the studies on performance in football referees have been conducted with men. However, gender differences appear in performance, so further research in females may clarify whether these female referees could improve their performance and thus approach the performance of male football referees. Future studies should explore the impact of female referees’ training sessions, as well as investigate strategies to prevent injuries.

## Figures and Tables

**Figure 1 ijerph-19-10720-f001:**
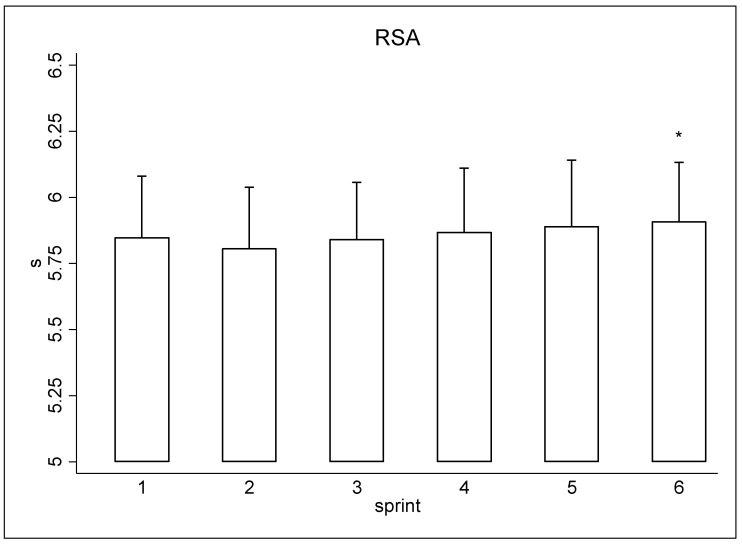
Performance of the RSA test (6 × 40 m) for female field referees. * Significant differences between sprints (*p* < 0.05).

**Figure 2 ijerph-19-10720-f002:**
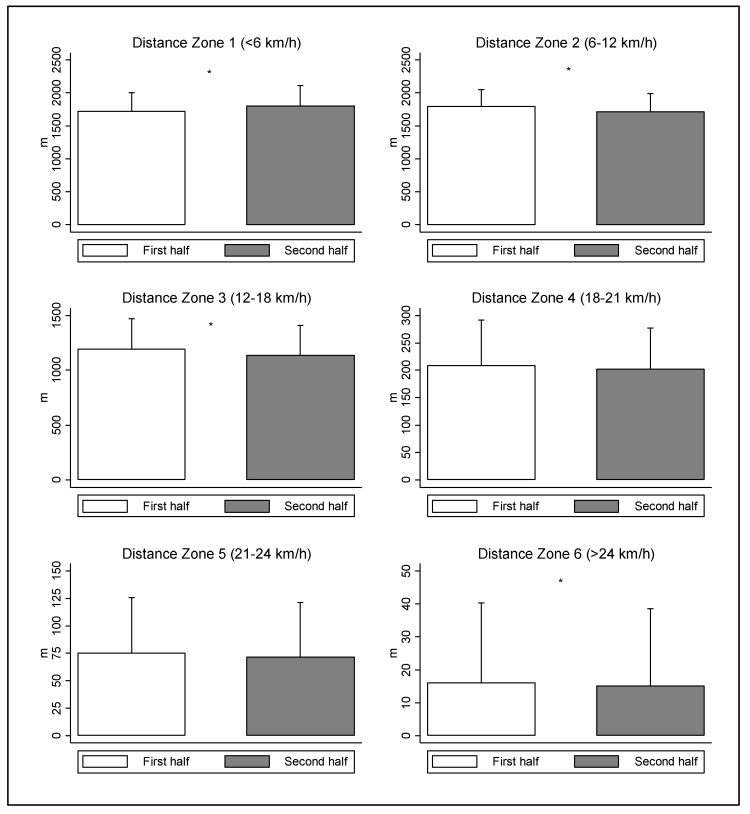
Distance covered in elite football matches by female field referees in different speed zones. * Significant differences between first and second half (*p* < 0.05).

**Table 1 ijerph-19-10720-t001:** External load variables related to distance covered in elite football matches by female field referees.

Variables	First Half	Second Half	Sig. (*p*)	ES	95 % CI	
Total distance (m)	5007 ± 404	4938 ± 392 *	0.01	0.17	16.41	122.56
Explosive distance (m)	565 ± 120	522 ± 125 *	0.00	0.36	32.82	54.79
HIBD (m)	53.7 ± 22.5	76.7 ± 41.7 *	0.00	0.72	−27.80	−18.12
Sprinting distance (m)	34.2 ± 43.4	33.3 ± 41.9	0.76	0.02	−4.91	6.73
HSRA (n)	9.0 ± 5.4	8.5 ± 5.1	0.11	0.10	−0.12	1.13
HSRD (m)	156.3 ± 99.2	152.0 ± 96.2	0.50	0.04	−8.22	16.81
Maximal Speed (km/h)	25.5 ± 2.2	25.1 ± 2.1 *	0.01	0.20	0.08	0.76

* Significant differences between first and second half in female field referees (*p* < 0.05); ES: effect size; CI: confidence interval; HIBD: High Intensity Break Distance; HSRA: High-Speed Running Actions; HSRD: High-Speed Running Distance.

**Table 2 ijerph-19-10720-t002:** Relationship between acceleration and deceleration demands of female field referees in match play.

Variables	First Half	Second Half	Sig. (*p*)	ES	95% CI	
ACC (n)	1312.0 ± 112.6	1382.4 ± 119.0 *	0.00	0.61	−85.22	−55.73
DEC (n)	1314.8 ± 111.4	1385.9 ± 118.2 *	0.00	0.62	−85.95	−56.26
DEC_MAX_ (m/s^2^)	−4.71 ± 0.83	−4.46 ± 0.67 *	0.00	0.33	−0.39	−0.10
ACC_MAX_ (m/s^2^)	4.13 ± 0.75	3.96 ± 0.71 *	0.02	0.23	0.03	0.31
ACC/DEC ratio	−5.98 ± 6.16	−4.76 ± 6.42 *	0.01	0.19	−2.15	−0.29
ACC Z1 (n)	1014.2 ± 115.7	1099.4 ± 126.9 *	0.00	0.70	−99.18	−71.37
ACC Z2 (n)	223.4 ± 38.5	215.5 ± 40.0 *	0.00	0.20	3.71	12.07
ACC Z3 (n)	63.6 ± 17.4	58.2 ± 18.5 *	0.00	0.30	3.53	7.21
ACC Z4 (n)	10.8 ± 7.2	9.3 ± 7.3 *	0.00	0.21	0.82	2.26
ACC Z1 (m)	1057.6 ± 143.3	1089.5 ± 156.8 *	0.00	0.21	−60.08	−18.46
ACC Z2 (m)	736.7 ± 97.7	734.3 ± 106.1 *	0.00	0.02	12.97	54.26
ACC Z3 (m)	286.7 ± 76.3	259.1 ± 74.6 *	0.00	0.37	19.66	49.00
ACC Z4 (m)	85.4 ± 44.2	70.2 ± 44.8 *	0.00	0.34	10.54	25.26
DEC Z1 (n)	1022.2 ± 113.0	1106.2 ± 124.5 *	0.00	0.71	−97.88	−70.08
DEC Z2 (n)	213.7 ± 36.3	210.2 ± 38.5	0.09	0.09	−0.61	7.60
DEC Z3 (n)	62.1 ± 16.6	55.5 ± 17.5 *	0.00	0.39	4.75	8.48
DEC Z4 (n)	16.8 ± 7.9	14.1 ± 8.9 *	0.00	0.33	1.84	3.67
DEC Z1 (m)	1126.6 ± 154.8	1165.8 ± 169.6 *	0.00	0.24	−51.62	−12.32
DEC Z2 (m)	1150.5 ± 173.7	1116.9 ± 177.6	0.74	0.19	−11.53	16.30
DEC Z3 (m)	495.3 ± 129.3	460.9 ± 132.6 *	0.00	0.26	18.30	36.88
DEC Z4 (m)	101.8 ± 72.4	83.9 ± 64.5 *	0.00	0.26	9.92	20.50

* Significant differences between first and second half in female elite referees (*p* < 0.05); CI: confidence interval; ES: effect size; ACC: Accelerations; DEC: Decelerations.

**Table 3 ijerph-19-10720-t003:** Time speed in the RSA test and distance in Yo-Yo Intermittent Recovery Test Level 1 in female field referees’ fitness tests.

Variables	RSA_TT_	RSA_BEST_	% RSA_BEST_	% RSA_DIFF_	Yo-Yo Intermittent Test
Total distance (m)	−0.229	−0.220	0.018	0.141	0.442
Explosive distance (m)	−0.346	−0.307	−0.105	−0.054	0.465 *
HIBD (m)	−0.365	−0.333	−0.074	−0.013	0.508 *
Sprint distance (m)	−0.656 **	−0.668 **	0.267	0.317	0.279
HSRA (n)	−0.332	−0.297	−0.097	0.072	0.306
HSRD (m)	−0.503 *	−0.489 *	0.075	0.203	0.329
Maximal Speed (km/h)	−0.632 **	−0.634 **	0.206	0.206	0.207
ACC (n)	0.550 *	0.513 *	0.047	0.022	−0.443
DEC (n)	0.526 *	0.490 *	0.052	0.030	−0.424
ACC Z4 (n)	−0.310	−0.275	−0.094	0.048	0.531 *
DEC Z4 (n)	−0.447 *	−0.412	−0.061	−0.021	0.462 *
ACC Z4 (m)	−0.393	−0.359	−0.066	0.121	0.535 *
DEC Z4 (m)	−0.509 *	−0.490 *	0.043	0.100	0.367

* Significant correlation between match variables and fitness tests in female field referees (*p* < 0.05). ** Significant correlation between match variables and fitness tests in female field referees (*p* < 0.01). HIBD: High-Intensity Break Distance; HSRA: High-Speed Running Actions; HSRD: High-Speed Running Distance; RSA: Repeat Sprint Ability Test; % RSA_DIFF_: percentage of difference between the best and worst sprint during the RSA test; ACC: Accelerations; DEC: Decelerations.

## Data Availability

Data is private and can only be shared under a data processing contract.

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
