# Peer review of "Association between Fitness Level and Physical Match Demands of Professional Female Football Referees"

_ijerph, 2022, doi:10.3390/ijerph191710720_

Round 1

Reviewer 1 Report

Comments to the Author

The manuscript titled "Association between Physical Performance and Match-Play Activities of Elite Women Football Referees" analyzes the match performance and fitness levels of elite women football referees for the first and second half as a function of playing time. However, there are several points that require further clarity;

1- Page 1, Lines 16 and 21: full words should be written before abbreviations. For example, acceleration (ACC). Please apply in others.

2- Page 1, Line 19-23: Please support these sentences with your numerical outputs. For example ACC (first half vs second half: 1312.0 ± 112.6 vs 1382.4 ± 119.0, etc.)...

3- Page 1, Line 30-35: The main purpose of the study was not written to emphasize the use, benefits or importance of the GPS system, so this paragraph is unnecessary. please remove this paragraph

4- Page 1, Line 30-60: I have some concerns about the introduction. I think it should be further developed and made more comprehensive from the general to the specific.

5- Page 2, Line 63-64: Sample size analysis is not specified, power analysis is not done.

6- Page 2, Line 70-72: Did they follow the same training program? and please explain the contents of strength, endurance, and speed.

7- Page 2: Physical performance may differ at different times of the day (circadian rhythm). So I wonder if all the analyzed matches were between the same times? Please explain.

8- Page 2, Line 81-84: Please provide more details. For example, how many meters apart were the photocells placed, or which feet were used at the beginning or at the turns, etc.? It is important to note that the test is standardized.

9- Page 2, Line 87-88: Please provide more details.

10- Page 3, Line 117-118: What is the RSAE? Why did you use the Paired samples t Test to analyses the difference between the time obtained in the second to sixth sprint of RSAE and the first one. Why didn't you use the repeated measure ANOVA for six different time? Please explain.

11- Page 5, Line 149-150: What is the mean‘’* Schema 0’’. Please explain.

12- Page 4-5, Lines 131 and 148: The numbers on the y lines in the figures have shifted. It should be revised.

13- Page 7, Line 196-197: Please rephrase. The sentence is ambiguous.

14- Page 7, Line 202: Please revise as ‘’the’’

15- Page 7, Line 198-200: The subject of the study does not discuss the ‘’valuable’’ of the tests, so this sentence can be removed. Please discuss your results from the RSA test and Yo-Yo Intermittent Recovery.

16- Page 7-8, Line 191-247: In this section, "correlations between the fitness levels and the physical demands during match play competition in the female football referees" were not discussed comprehensively. The title of the study is " Association between Physical Performance and Match-Play Activities of Elite Women Football Referees", so need to discuss the correlations in detail in this section.

GENERAL COMMENTS:

1. The manuscript requires language improvement.

2. The topic is important but especially the discussion section should be improved

significantly. Literature review is nonadequacy.

3. Abstract should be re-edited after changes made in the article.

4. There are errors with the use of abbreviations.

Author Response

Response to Reviewers’ comments

Association between Physical Performance and Match-Play Activities of Professional Female Football Referees

Author's response: We sincerely thank the expert Reviewers and the International Journal of Environmental Research and Public Health for their helpful and constructive comments, and for carefully reviewing the manuscript. In this rebuttal letter, we have addressed all the points raised by the Reviewers and we have listed all the corresponding changes performed throughout the manuscript, highlighted in red. We believe that the comments raised by the Reviewers and the resulting text revisions have significantly improved the quality of this manuscript.

Reviewer 2 Report

This paper aims to determine the physical load during football matches in female elite referees and its relationship with their performance in physical tests. I think the paper deals with an important and understudied topic (performance in female referees), the methodology is good and the level of participants is very high (which is not easy to obtain), so I congratulate the authors for this.

I have some minor comments which came to my mind when reading the paper. It is not necessary that authors change the paper to adapt to all of them, but only to reply to me if they decide to maintain the current form. Furthermore, I hope some of my comments are useful to improve the quality of the paper.

Page 1, line 2: The expression “match-play activities” is used in the title but later in the paper is not defined and scarcely used. I suggest to change it for another expression in the title or to define it in the introduction and methods, so to include it along all the paper sections.

Page 1, line 13 and along all the paper: I am not sure if "demands" is the optimal word to be used in this context (unless the literature in the field had consistently used it). "Demand" means request, petition, solicitation, claim, need, order, inquire,… So, it implies what the referee should do (i.e.: she should run to be near the ball) but you are measuring what they actually do, what is not the same. I think you are measuring "physical performance", "physical load", "distance", "speed", "acceleration",,... but not "demands". Please, change the word (expression) in all the paper or justify to me in your response the use of this expression (e.g., if the literature in this field uses it).

Page 1, line 21: I suggest to avoid acronyms in the abstract (YYIR1, ACC, DEC, Z1,…) or, at least, to define it when you use it for the first time as you did with GPS.

Page 1, introduction: I suggest to start the introduction focusing on the importance of physical performance in football referees instead the GPS technology.

Page 1, line 30: In the first sentence of the paper, you stated “clubs are investing financial and time…” which is true, but as clubs do no usually invest resources in referees. So, I would talk about federations, organizations, etc., but not clubs. Furthermore, this paper is not related to “competitive advantage”, so I think the first sentence should be more closely related to the paper content.

Page 1, line 40: The sentence "but there are not many studies on the relationship between women and football" sounds too general (scarce content). Could you be more specific? It is not easy to know what authors mean.

Page 1 and 2: In general, I think the introduction can be improved, so feel free of changing the structure and content if you want.  

Page 2, line 63-79: I miss to know the range of matches that each referee judged (for example, each participant refereed between 3 and 15 of this 187 matches). I can imagine that not all the referees judged the same number of matches.

Page 2, line 82: A really minor issue: As far as I understand, you only use two fitness tests (RSA and Yo-Yo). I think you should not use the expression of "battery of fitness test" but “two fitness tests” since “battery” usually implies more tests (endurance, power, flexibility,…).

Page 2, line 84: I think RSA-mean and RSA-TT are measuring the same construct (RSA-mean = RSA-TT / 6), are they? In fact, the two first columns of table 3 have the same values. I suggest to delete one of the two to avoid double information. If you agree, please delete all the information of one of them along the paper (abstract, methods, results,…).

Page 2, line 88: I suggest to include the name of the variable (physiological concept) that tests are measuring. For example, the Yo-Yo is a measure of aerobic capacity. The sentences could be “the female football referees performed a Yo-Yo Intermittent Recovery Test Level 1 to determine their aerobic capacity”. I think it would be more meaningful.

Page 3, line 121: Please, report the name of the variables consistently along the paper. For example, “performance test” are called “fitness test” just before in the same section. To be consistent will avoid confusion among readers.

Page 4, line 131: To be honest, I do not think the figure 1 represents such important data to merit a figure. If you agree you can delete it and include a more meaningful figure in other part of the paper. For example, I think a figure that reports the regression of some values from fitness test (e.g. yo-yo test in the X axis vs total distance during the matches in the Y axis) would be nice to see at first glance if those referees who are fitter in the fitness tests also cover greater distances during the matches.

Page 4, line 135: Again, the expression "physical demands" sounds inaccurate for me. Is really the second half of the match "less demanding" than the first half? Or the demand is similar but referees cover less distance because they are tired?

Page 4-5: In methods you mention the ES level (trivial, small, medium and large) but later in results/discussion I cannot see if you use this. Please comment on it in the text in results.

Page 6, line 174: Maybe the word "correlation" or "relationship" should be included in the table caption to better explain the content of the table.

Page 6-7, table 3: I suggest to delete one of the two first columns since they have the same values.

Page 7, line 193: I suggest to change the expression “higher results” for a more meaningful expression that will help understanding.  

Page 7, line 197: I think this sentence should be re-written: “On the other hand, the results in the fitness test and the physical demands in the match, therefore, it is very important to evaluate the fitness level of the female football referees”

Page 7, line 202: I think “th match” should be “the match”.

Page 8, line 209: I suggest to change the expression “lower results” for a more meaningful expression that will help understanding. Maybe “lower physical performance”.

I congratulate the authors for their effort in this research. I hope my comments are helpful.

Author Response

(The authors gave the same response as above.)
